# In-Mould OCT Sensors Combined with Piezo-Actuated Positioning Devices for Compensating for Displacement in Injection Overmoulding of Optoelectronic Parts

**DOI:** 10.3390/s23063242

**Published:** 2023-03-19

**Authors:** Günther Hannesschläger, Martin Schwarze, Elisabeth Leiss-Holzinger, Christian Rankl

**Affiliations:** 1Research Center for Non-Destructive Testing GmbH (RECENDT), Science Park 2, Altenberger Str. 69, 4040 Linz, Austria; 2Fraunhofer Institute for Machine Tools and Forming Technology, Reichenhainer Straße 88, 09126 Chemnitz, Germany

**Keywords:** optical coherence tomography, micro injection moulding, optoelectronics, in-mould sensor, in situ measurement, image processing, mechatronic actuator

## Abstract

When overmoulding optoelectronic devices with optical elements, precise alignment of the overmoulded part and the mould is of great importance. However, mould-integrated positioning sensors and actuators are not yet available as standard components. As a solution, we present a mould-integrated optical coherence tomography (OCT) device that is combined with a piezo-driven mechatronic actuator, which is capable of performing the necessary displacement correction. Because of the complex geometric structure optoelectronic devices may have, a 3D imaging method was preferable, so OCT was chosen. It is shown that the overall concept leads to sufficient alignment accuracy and, apart from compensating for the in-plane position error, provides valuable additional information about the sample both before and after the injection process. The increased alignment accuracy leads to better energy efficiency, improved overall performance and less scrap parts, and thus even a zero-waste production process might be feasible.

## 1. Introduction

Micro-injection moulding has become a widespread and well-established method for mass manufacturing small plastic parts of various materials [1,2]. One increasingly important application is micro-injection moulding of optical parts, such as lenses and diffraction gratings [3,4]. Usually, the injection-moulded parts need to be assembled with optoelectronic parts in a dedicated production step. This leads to various disadvantages. From the production point of view, it increases the overall process complexity and is time-consuming. From the quality point of view, it has the issue of inaccuracies due to the tolerances of the part itself and the assembly process, and it suffers from energy losses in the air gap between the LED and optical lens. The back reflection at each air/polymer interface is approximately 4% [5], which leads to a significant loss in efficiency, expecially for optical systems with more than one element. Direct overmoulding of the optoelectronic element is an innovative improvement of this process, since it makes the assembly step obsolete, gets rid of the air gap and makes the whole part more compact. However, to ensure high product quality and a low number of deficient parts, direct overmoulding requires precise alignment of the optoelectronic part with respect to the mould before the injection process (see Figure 1). In order to achieve this alignment, a sensor is needed, which determines the positioning error, as well as an actuator, which is capable of correcting the measured error. Solutions to both tasks are presented in this publication, customized for the particular use case of a fiber optic transceiver (FOT).

In this particular case, the positioning inaccuracies that have to be dealt with have their origin in the placement tolerances of the LED on a metal lead frame and the overall manufacturing tolerances. Both can be solved with our approach. As can be seen in Table 1, the maximum positioning error using the conventional method is 55 µm.

To compensate for the measured displacement, a mould-integrated mechatronic actuator was developed [6]. It comprises a compliant kinematic mechanism which is driven by four piezo-actuators. This mechanism is capable of correcting 300 µm of displacement in both lateral directions and 1.8 degrees rotationally.

In this work, we develop a closed-loop system by controlling the actuator based on sensor information acquired by an optical coherence tomography (OCT) sensor. The performed in-mould measurements show that OCT is a valid in-mould measurement tool that can be used for in-plane positioning. The system allows reducing the remaining displacement down to 1 µm. In addition, OCT provides even more data about the sample, so additional applications are possible, such as out-of-plane displacement detection and post-injection quality control.

## 2. Methods: Optical Coherence Tomography

Optical coherence tomography has been around since the early 1990s, focusing at first on ophthalmic applications [7,8]. Over the years, technological advances have increased the speed [9], resolution [10], wavelength range [11,12,13] and sensitivity [14] of OCT set-ups, which opened up many new applications for this fast, non-contact, non-destructive and robust measurement technology, such as pharmaceutical tablet coatings [15], multi-layer polymer films [16] and polymer coextrusion [17], as well as many more [18]. Being well established as a lab measurement tool both in medical and industrial fields, particular effort is needed when using it as an in-line measurement tool in the production process.

### 2.1. Optical Coherence Tomography System

OCT is an interferometer-based method which usually provides depth scans of the sample under testing. Due to reasons of speed and sensitivity, we decided to use a so-called spectral domain OCT set-up (SD-OCT). The principle of the SD-OCT is shown in Figure 2. The OCT set-up’s specifications had to be chosen according to the requirements for measuring the FOT. For that reason, we decided to use a custom-built OCT device. Table 1 shows the requirements and specifications. In total, 10 different FOTs were measured: 5 in an overmoulded state and 5 in a non-overmoulded state. The set-up was designed according to the well-known sample specifications and tolerances, so the number of samples was sufficient. The depth and lateral scanning ranges were chosen to comfortably image the region of interest on the sample, whereas a wavelength range of around 840 nm was selected because of the high transparency of the lens material and to achieve a better resolution.

Since the device should be capable of in-mould measurements, the probe head had to meet very specific requirements. First, the probe head’s thermal expansion coefficient had to match the mould material, so it was manufactured using steel. Second, since the sample had to be measured from above, the injection channel had to be redesigned to not overlap with the optical path. Third, since the mould consists of a fixed (lower) and moving (upper) part, the probe head was also split into a fixed and moveable part. The sample itself, the mechatronic actuator and the main part of the probe head were mounted to the fixed part. Only the reference path had to pass through the moveable mould part. Figure 3 shows two cross-sections through the resulting probe head design. Figure 4 shows the custom-built OCT device and the probe head mounted to a 3D-printed mock-up mould. This was very useful for adjusting the OCT device and performing test measurements to ensure the actual in-mould measurements in combination with the displacement correction device would work smoothly.

### 2.2. Positioning Error Calculation

As explained in Section 2.1, the basic measurement of an SD-OCT system is a depth scan, also called an A-scan. As the beam is scanned laterally along a line, a B-scan (or cross-section) is provided. Several adjacent B-scans then form a stack of B-scans, or a volume scan. In our case, each A-scan had 1024 pixels, each B-scan had 1000 A-scans, and 800 B-scans were recorded in one measurement. Since the sample surface lies in a plane perpendicular to the OCT beam, several en face scans were selected and summed up along the depth direction as input for the displacement calculation algorithm. This approach is advantageous compared with a traditional camera-based system. In a camera image, not only would the light-emitting diode (LED) be visible but also every surrounding structure. With OCT, it is easy to precisely cut out the LED surface without any influence from the die or the lead frame. Figure 5 shows both the construction of the volume scan and the extraction of the en face images, which are in a plane perpendicular to the OCT beam.

The displacement values were determined using two different procedures. First, the cross-correlation between a template and the en face image was calculated. The template represents the zero-displacement case, and the en face image contains the image of the misaligned sample. While the template may be based on a real measurement, distortions and imperfections should be removed to make it equally applicable to all samples under testing. Since the template plays a major role in the correction procedure, for finding the perfect template, it would be necessary to produce test samples (FOTs) that could be examined regarding the beam’s shape and efficiency. The result of the first step is displacement in the x and y directions. Second, the en face image was processed with canny edge detection and a Hough transform-based line detection algorithm, which resulted in angular displacement between the template and en-face image. Figure 6 shows these steps applied to a test image (top row) and real data (bottom row).

In addition to the procedure mentioned in [19], the angular correction was refined in order to also match strong angular misalignment and to make the procedure robust against low-quality images. Using Figure 6f as an example, it works as follows. The straight line Hough transform detects three straight lines (green) that exceed the minimum line length. Using the Hough line length as a weight, the weighted average of the straight lines gives the total angular orientation of the misaligned sample. A significant angular correction may introduce an additional linear shift, and therefore a second linear displacement correction is applied. The total Euclidean transformation matrix is the product of these calibration-corrected shift and rotation operations. In the end, this transformation matrix is sent to the actuator.

## 3. Methods: Mechatronic Actuator

The device for compensating for the measured positioning error had to meet a couple of requirements. First, it should be able to move ±300 μm on both the *x* and *y* axes and rotate ±1.8 degrees in the plane while still meeting the requirements of a movement accuracy of 0.1 µm translationally and 0.6° rotationally. Second, it needs to work at a temperature of 80 °C. Third, a compact design is necessary to fit into the mould core of a micro injection moulding machine. Consequently, a piezo-actuated three-degrees-of-freedom (DOF) mechanism with flexible joints was developed. The kinematic structure and design process was described in great detail by Rentzsch et al. [6].

### 3.1. Actuator Design

The final design of the actuator included a compliant mechanism that amplified the movement of the piezo elements. Two piezos were used for translation in the x direction which were located at the bottom, and two were used for translation in the y direction and the rotational movement around the (vertical) z axis, which were located at the top. Finally a plate for mounting the FOT was placed on top. Figure 7 shows the top surface of the actuator, mounted inside the lower part of the mould. The motion control system was running on a CompactRio module from National Instruments (NI).

### 3.2. Combined Set-Up

The mechatronic actuator and the OCT system were combined at the Fraunhofer Institute for Machine Tools and Forming Technology in Chemnitz, Germany. Due to careful preparations, all components fit together well. In order to allow the actuator to move, a 1 mm gap between the upper and lower mould parts was maintained during measurement, which was closed after the position correction. A server/client structure based on the TCP/IP protocol was chosen for communication between the OCT and actuator systems. The server, running on the OCT system, is waiting for the client’s signal to start the OCT measurement, tells the OCT software to do so and returns the displacement values to the client. All tests and measurements were performed under laboratory and workshop conditions.

## 4. OCT Measurements and Results

For developing the algorithm, extensive test measurements on non-overmoulded and overmoulded parts were made. First, it was necessary to find out which features of the sample were visible in the OCT scans and then if and how information could be extracted from them. Figure 8 shows a typical B-scan of a non-overmoulded FOT with the most important features pointed out. In this case, only the FOT surface is the part of the OCT signal that is used for calculating the displacement information. Based on the fact that the FOT surface appeared to be flat and homogeneous in all test measurements, the algorithm described in Section 2.2 was developed. All tests and measurements were performed under laboratory or workshop conditions: room temperature, no air conditioning, dry, standard table or metal workbench and no vibration dampening.

### 4.1. Test Measurements with Mock-Up Mould

In-house test measurements with the mock-up mould were important to adjust the probe head, particularly the objective lens position in the sample path. The focus point of this lens was laid close to the FOT’s surface position to obtain good image quality. The samples showed distinct differences in reflectivity and position between each other, as can be seen in Figure 9. Therefore, the algorithm needed to be robust enough to not favor one of these samples. Then, everything was prepared for combining the OCT sensor and the mechatronic actuator. Additionally, measurements of the overmoulded samples were made, which will be discussed later. A basic measuring cycle comprises placing the FOT into the mould, closing the mould, performing the OCT measurement and taking the FOT out of the mould. This was performed multiple times in a row with the same sample. Since the subsequent measurements were all perfectly the same, we could eliminate the possibility of the OCT device causing additional displacement. Finally, we had to find a suitable scanning range for the galvanometer mirrors. The test measurements showed that for the FOTs, a lateral scanning range of 1 mm in the x direction and 0.8 mm in the y direction was sufficient, which also led to a pixel-to-µm ratio of roughly 1 µm/px in both directions. This scanning range was also used in the combined set-up trials described below.

### 4.2. Positioning Accuracy Test

To test the positioning accuracy, the actuator movements were measured with a Renishaw laser interferometer with a maximum measuring accuracy of 0.1 µm. Figure 10 shows the complete measurement set-up, including a beamsplitter and a reflector that was mounted on the actuator. The tests were carried out according to the Association of German Engineers’ guidelines VDI 3441 and ISO 230 (test code for machine tools), which included multiple approaches to each actuator position from both directions. The results are displayed in Figure 11. Two problems could be identified: drift in the x direction and a parasitic *x*-axis movement when the *y* axis was operated. The first one was caused by sensor noise on the *x* axis but did not pose a significant problem, because the OCT measurement and position correction were much faster than this drift. The latter one was caused by manufacturing tolerances and had to be dealt with, since it could also be detected in the OCT measurements, as described later. However, this can be eliminated by proper calibration.

### 4.3. Calibration Measurements

After combining the OCT system with the actuator, a calibration procedure was performed to find out the elements of the algorithm’s transformation matrix. First, a template was created by making an OCT scan at the actuator’s zero position. This was used as a reference. Then, well-defined single-axis movements of the actuator were induced, namely $x_{set}$ *or* $y_{set}$ in micrometers, each followed by an OCT displacement measurement in relation to the reference image, namely $x_{OCT}$ *and* $y_{OCT}$ in pixels. According to Equation (Equation 1), we determined the elements of the matrix *A*. After this calibration, the inverse of this matrix, along with the displacement detected by OCT, was used to calculate the corrective movements for the actuator as shown in Equation (Equation 2). Because of the parasitic movement described in Section 4.2, *A* is not a diagonal matrix. Figure 12 shows the results of this calibration procedure:(1)$${\left(\begin{array}{c} x\\ y \end{array}\right)}_{\mathrm{OCT}}=A{\left(\begin{array}{c} x\\ y \end{array}\right)}_{\mathrm{set}}=\left(\begin{array}{cc} a_{11} & a_{12}\\ a_{21} & a_{22} \end{array}\right){\left(\begin{array}{c} x\\ y \end{array}\right)}_{\mathrm{set}}$$
(2)$${\left(\begin{array}{c} x\\ y \end{array}\right)}_{\mathrm{actuator}}=A^{-1}{\left(\begin{array}{c} x\\ y \end{array}\right)}_{\mathrm{OCT}}=\left(\begin{array}{cc} 1.18 & 0.21\\ 0 & 1.39 \end{array}\right){\left(\begin{array}{c} x\\ y \end{array}\right)}_{\mathrm{OCT}}$$

In the ideal case, the matrix would be a diagonal matrix. Unfortunately, we detected some parasitic movement from the y axis in the x direction, very likely because of some imperfections in the compliant mechanism. Since it was reproducible, it could be taken into account when calculating the displacement compensation values. Another issue was the drift we detected in the x axis, which was caused by one of the actuator’s position sensors. Since this drift was not fast (taking several minutes), it would not cause problems in the normal process of displacement correction, which takes a maximum of 20 s. The similarity of these results to the laser interferometer measurements presented above shows that the OCT measurements are trustworthy.

### 4.4. Measurements for Displacement Correction

After completing the calibration process, we needed to prove that the measurement system and the actuator were able to correct the initial positioning error of an FOT. The precision and repeatability were tested as follows. To test the performance regarding precision, arbitrary initial displacements between 20 µm and 200 µm were induced with the actuator. Then, the displacement was measured, sent to the actuator control and corrected by the actuator. In order to see if the remaining deviation was small enough, this measurement and correction procedure was performed multiple times. Table 2 shows the calculated deviations before and after each correction cycle. These measurements gave very pleasing results, with the remaining displacement being smaller than 1 pixel or 1 µm.

The repeatability and reliability of the process were examined by a similar procedure. The only difference was that before the next displacement correction cycle, the sample was unmounted, and the same sample was mounted again. This helped us find out if the mounting step itself caused any additional displacement and if the displacement correction was reliably giving the same results. Since these tests did not show any detectable deviations, it was proven that the proposed method is capable of correcting the displacement error with high accuracy.

### 4.5. Additional Value of the OCT Data

OCT can do a lot more than just replace a camera for measuring in-plane displacement. The 3D data set provided by OCT includes information about the height and tilt of the LED, position of the die on which the LED is mounted, the lead frame, glue residues and the condition of the bonding wires (see Figure 13). Moreover, OCT can be used for post-injection measurements as a tool for quality assurance (see Figure 14). Naturally, dedicated algorithms are necessary to gather information about these features.

## 5. Discussion and Conclusions

In this contribution, it was shown that OCT is a valid tool to measure in-plane positioning errors, which can be used as input for a position correction device. The remaining displacement could be reduced to ±1 µm with high repeatability. In the final product, the displacement correction led to higher optical efficiency and a better beam shape and therefore fewer scrap parts. One displacement correction cycle took roughly 20 s. There is potential for reducing this cycle time, mainly by reducing the number of B-scans and scanning only the relevant parts of the sample.

The novelty of this system lies not so much in the OCT measurement itself but in the application as an in-mould sensor, which has not been attempted before, and the combination with the mechatronic actuator. It was necessary to develop a customized probe head design with a movable and fixed part and to establish a new measurement and correction procedure. In addition, other potentially useful features were pointed out that can be examined via OCT, such as the visibility of bonding wires, glue residues and out-of-plane displacement.

Along the development process, a couple of issues could be identified that have to be dealt with before actually implementing such a sensor into an injection moulding machine and applying it to an FOT production line. Some have to do with nonlinearities in the actuator movement and the displacement that is caused by closing the mould before the injection process. Another important challenge is the transparent sapphire-glass viewport, which cannot be manufactured yet with the negative shape of the lens inscribed into the inner surface of the viewport. Using a different viewport material, which is easier to machine, might be a useful approach to be able to manufacture such a structure. It is important to point out that such a viewport will strongly influence the measurement because the glass/air interface diffracts the OCT measuring beam. Thus, the algorithm for calculating the displacement needs to be adapted. On the other hand, the interpretation of the post-injection measurements will be easier, since the glass/air interface turns into a glass/polymer interface. As the refractive index of glass roughly matches the refractive index of polymer, the optical distortion caused by this interface will be very low. Thus, the appearance of the FOT in the OCT image will be close to the one with a flat viewport.

## Figures and Tables

**Figure 1 sensors-23-03242-f001:**
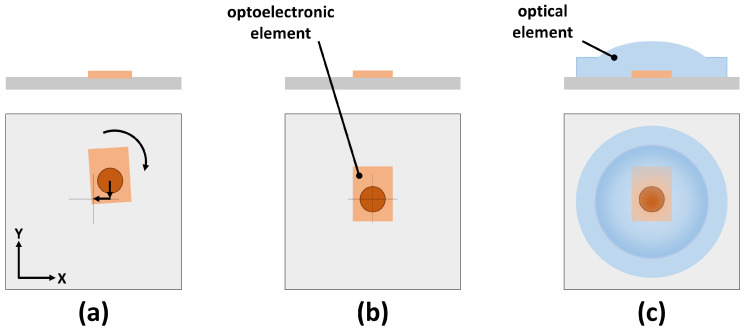
Displacement correction of optoelectronic element with regard to the overmoulded optical element. (**a**) In-plane displacement and two translational and one rotational dimensions. (**b**) Optoelectronic element after displacement correction. (**c**) Part after overmoulding.

**Figure 2 sensors-23-03242-f002:**
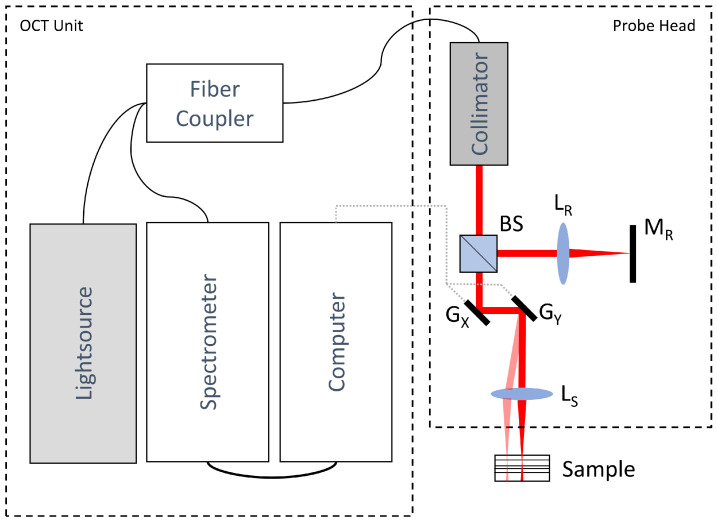
Principle of SD-OCT: Broadband light is sent through a fiber coupler to the probe head. Inside the probe head, the light is collimated and sent through a Michelson interferometer, which divides the collimated beam into sample and reference beams. The sample beam is focused and scanned over the sample under testing along two directions. The reflected light interferes with the reference beam. A spectrometer detects the so-called interferogram, which then is sent to the computer for further data processing. The parts included in the probe head are the beamsplitter (BS), reference path lens (L_R_), reference mirror (M_R_), galvano mirrors for the x and y directions (G_X_, G_Y_) abd focusing lens (L_S_).

**Figure 3 sensors-23-03242-f003:**
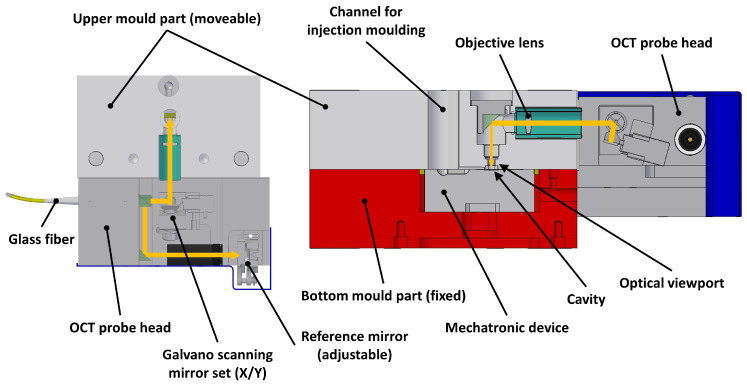
Cross-sectional views of the probe head. The beam path is indicated by the orange line, with a view from the top (**left**) and a view from the side (**right**). The optical elements that are inside the movable mould part are the objective lens, right angle prism mirror and sapphire glass viewport.

**Figure 4 sensors-23-03242-f004:**
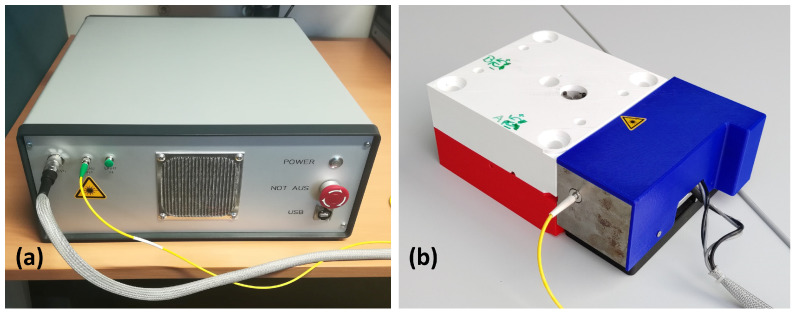
(**a**) Front view of the OCT device. All components except the probe head are built into a 19-inch housing. (**b**) Probe head (with a blue 3D-printed cover) mounted to the mock-up mould (fixed part in red and movable part in white).

**Figure 5 sensors-23-03242-f005:**
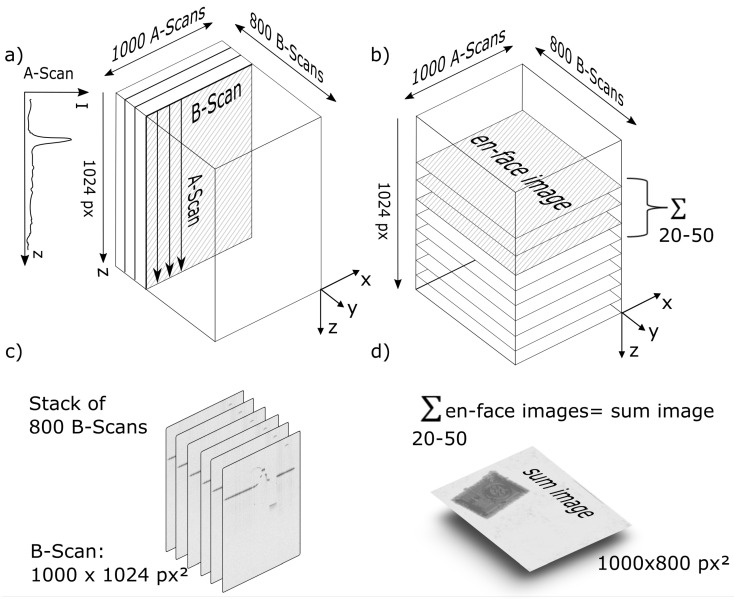
OCT scanning scheme. (**a**) A-scan intensity of 1024 pixels. Here, 1000 consecutive A-scans build a B-scan, which represents a cross-sectional image of the sample, while 800 B-scans form a volume scan, representing a 3D data cube of the sample. (**b**) From this 3D data cube, 1024 en face images can be extracted. For a noise-reduced en face sum image, 20–50 en face images are summed up. This sum image resembles a top view of the sample. (**c**) For sample characterization and detection of defects, B-scans are evaluated. (**d**) For misalignment evaluation, an en face sum image is used.

**Figure 6 sensors-23-03242-f006:**
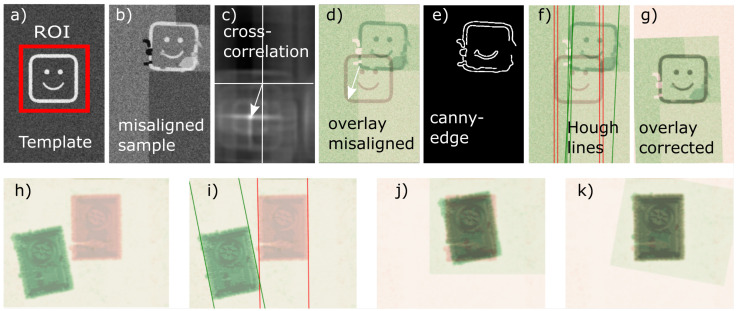
(**a**–**g**) Image-processing steps shown on test images. (**a**) Template: the structure of interest is in correct position and is selected as region of interest (ROI). (**b**) Misaligned sample: Test image containing the structure of interest at a different position and a variety of image distortions (e.g., noise or bad pixels). (**c**) Cross-correlation of the ROI with the misaligned sample. The white arrow indicates the in-plane displacement. (**d**) Overlay of the template and the misaligned sample. The white arrow again indicates the displacement. (**e**) Edge detection with canny edge algorithm on the misaligned sample. This is also applied to the template. (**f**) Hough line detection is applied to both edge images, with the results displayed as an overlay image. These are used for calculating the angular displacement. (**g**) Corrections in terms of linear and angular displacement lead to a perfect overlay of template and previously misaligned sample. (**h**–**k**) Application on real measured data. (**h**) Overlay of template (red) and misaligned sample (green). (**i**) Angular misalignment detected by Hough lines. (**j**) Correction of linear displacement only. (**k**) Linear and angular displacement correction, leading to perfect overlay.

**Figure 7 sensors-23-03242-f007:**
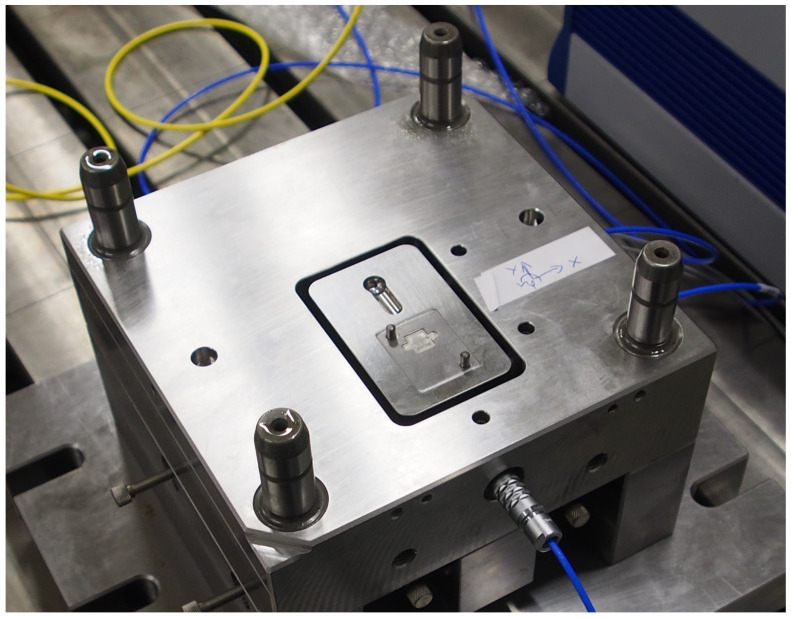
Top view of the mechatronic actuator mounted in the lower part of the mould. The upper part was removed for this image. A part of the injection channel can be seen as well as the dowel pins for mounting the FOT.

**Figure 8 sensors-23-03242-f008:**
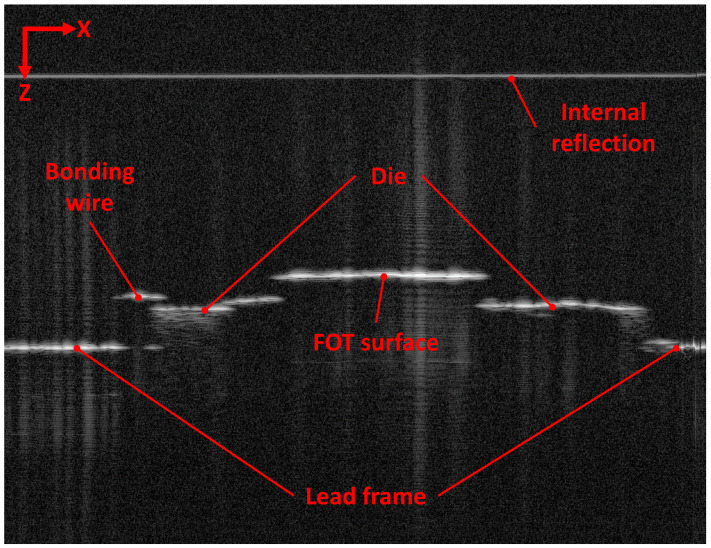
Typical B-scan image of non-overmoulded FOT. Apart from the LED’s upper surface, several other features can be identified. This includes the die on which the LED is mounted, the lead frame and the bonding wires. Additionally, unwanted signals due to internal reflections in the beam path might occur. The algorithm has to make sure that these parasitic signals do not interfere with the desired signals.

**Figure 9 sensors-23-03242-f009:**
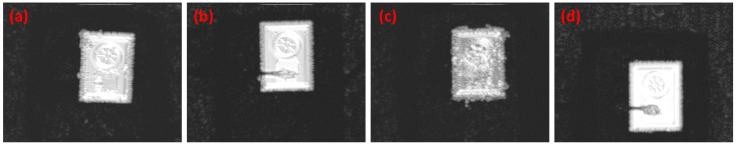
En face scans of four different non-overmoulded FOTs all at different positions. (**a**) Sample 4. (**b**) Sample 6. (**c**) Sample 7. (**d**) Sample 8.

**Figure 10 sensors-23-03242-f010:**
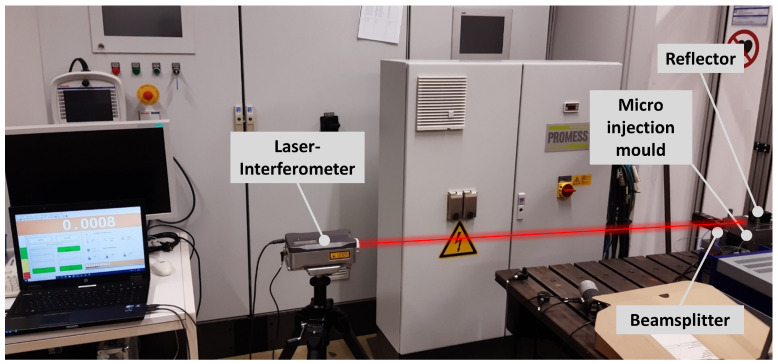
Set-up for calibration measurements of the mechatronic actuator. A mirror was mounted on the device, and its position was measured with a laser interferometer. By splitting the interferometer beam into two beams, the measurement could be made in the x and y directions without moving the device.

**Figure 11 sensors-23-03242-f011:**
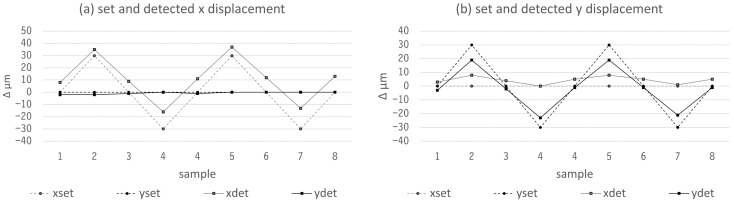
Positioning accuracy measurements for x and y axes. First, a displacement was induced by the controller (x set, y set), which then was measured by the laser interferometer (x det, y det). (**a**) Movement in x direction. The measurement shows a 10 µm offset and a small drift. (**b**) Movement in y direction. The measurement shows only a very small offset and no drift but a parasitic movement in the x direction.

**Figure 12 sensors-23-03242-f012:**
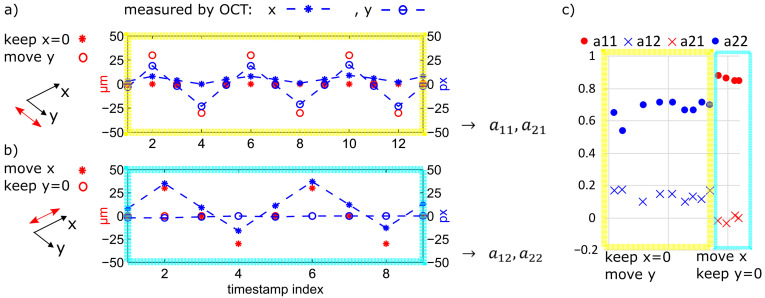
Translation calibration. (**a**) Here, x is held constant, and the y axis moves back and forth between two positions. The OCT sensor detects the y movement but also a parasitic movement in the x direction (blue asterisk line). (**b**) Here, y is held constant, and the x axis moves back and forth between two positions. The OCT image detected a slow drift in the x direction. The time between two measurement points was one minute. (**c**) Drift-corrected result of the matrix elements. The average of the results proved to be a sufficiently exact value.

**Figure 13 sensors-23-03242-f013:**
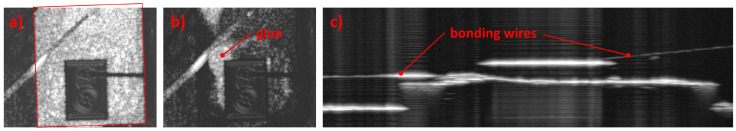
OCT images showing additional features. (**a**) En face sum image that shows how the LED was positioned on the die. (**b**) En face sum image showing the glue distribution between the LED and die. (**c**) B-scan sum image showing the positions and completeness of two bonding wires.

**Figure 14 sensors-23-03242-f014:**
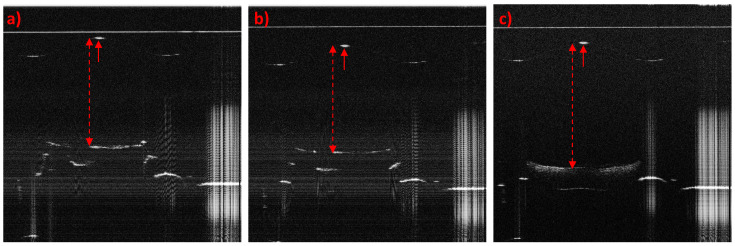
B-scans of overmoulded FOTs. The B-scans were taken through the center of the lens so that the lens apex was visible (small, red arrow). The center thickness of the moulded material can be calculated (red, dashed arrow). Because of the lens-shaped surface the OCT signal of the LED and die were distorted so their surface no longer appeared flat. Samples (**a**,**b**) are successfully overmoulded FOTs, and sample (**c**) is a scrap part, which can be seen in the significantly larger center thickness and the different appearance of the LED.

**Table 1 sensors-23-03242-t001:** Requirements for FOT position measurement and specifications of OCT set-up.

Specifications of FOT		
Circular positioning error of LED	<55 µm	
Position tolerance LED to optics	50 µm	x and y directions
Fabrication tolerance	20 µm	
Angular tolerance	0.06°	
FOT size in plane	<1 mm	squared
FOT height	<500 µm	not overmoulded
FOT height	2.5 mm	overmoulded
Refractive index of lens material	1.531	
**OCT Set-up Requirements**		
Center wavelength	840 nm	with 65 nm bandwith
Depth range	4.2 mm	
Lateral scanning range	5 mm	squared
Axial resolution	5 µm	
Lateral resolution	15 µm	
A-scan rate	50 kHz	

**Table 2 sensors-23-03242-t002:** Results of displacement correction trials. In test A, five measuring and correction steps were performed to prove that the mechanism was stable. All three tests show that after a maximum of two correction steps, the displacement was reduced to less than 1 micrometer.

		Displacement in px	
	Step	dx	dy
Test A	0	−175	199
	1	−20	−9
	2	1	−1
	3	0	0
	4	0	0
	5	0	0
Test B	0	−415	−32
	1	1	−2
	2	0	−1
Test C	0	98	63
	1	2	4
	2	1	−1

## Data Availability

Not applicable.

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
