# Peer review of "In-Mould OCT Sensors Combined with Piezo-Actuated Positioning Devices for Compensating for Displacement in Injection Overmoulding of Optoelectronic Parts"

_sensors, 2023, doi:10.3390/s23063242_

Round 1
Reviewer 1 Report
The authors developed a new mould-integrated optical coherence tomography device, and this device dealt with the problem of precise alignment of the mould. This research has a high value, and this device can be used in the future. I suggest that this paper shold be published after considering my comments. My comments are as below.
1. Please add the result comparison between using this method and not using this method.
2. Please add more contents to explain the Figure 6.
Reviewer 2 Report
The manuscript In-mould OCT-sensor combined with piezo-actuated positioning device for compensating displacement in injection overmoulding of optoelectronic parts by Günther Hannesschläger et al, it is a nice study that may be well accepted by a broad community. However, requires revision prior to take any decision. Please, see the comments below.
Titled: It is ok.
Abstract: this requires the identification of the reason why you select this sensing architecture for the application envisaged. Please clarify what the performances achieved and how they impact the present state of the art.
Introduction: Here when addressing the topic, it is relevant to clearly identify the existing bottlenecks and why the present study aims to solve the same. For instance, when looking for displacement, we could even use psd that may track properly what it is expected. The advantages of the proposed solution are missing.
Methods:
How many structures were tested? How reproducible ad reliable they are? What is the error associated when evaluating/testing? What are the environmental conditions in which the systems were tested? Did you notice ageing effects?
Measurements and results:
Well done but requires some more information. Can you elaborate more on the role of the configuration/architecture used in the set of performances achieved?
Discussion and Conclusions: Overall good, but too qualitative. Missing quantitative key performance indicators of the work performed.
Figures: Are OK.
References: require updated.
